

# Footfall patterns and stride parameters of Common hippopotamus (*Hippopotamus amphibius*) on land

John R. Hutchinson and Emily V. Pringle

Comparative Biomedical Sciences, Structure and Motion Laboratory, The Royal Veterinary College, North Mymms, Hertfordshire, United Kingdom

## ABSTRACT

Common hippopotamuses (hippos) are among the largest extant land mammals. They thus offer potential further insight into how giant body size on land influences locomotor patterns and abilities. Furthermore, as they have semi-aquatic habits and unusual morphology, they prompt important questions about how locomotion evolved in Hippopotamidae. However, basic information about how hippos move is limited and sometimes contradictory. We aimed to test if hippos trot at all speeds and if they ever use an aerial (suspended) phase, and to quantify how their locomotor patterns (footfalls and stride parameters) change with approximate speed. We surveyed videos available online and collected new video data from two zoo hippos in order to calculate the data needed to achieve our aims; gathering a sample of 169 strides from 32 hippos. No hippos studied used other than trotting (or near-trotting) footfall patterns, but at the fastest relative speeds hippos used brief aerial phases, apparently a new discovery. Hippos exhibit relatively greater athletic capacity than elephants in several ways, but perhaps not greater than rhinoceroses. Our data help form a baseline for assessing if other hippos use normal locomotion; relevant to clinical veterinary assessments of lameness; and for reconstructing the evolutionary biomechanics of hippo lineages.

# INTRODUCTION

Common hippopotamus (*Hippopotamus amphibius* Linnaeus 1758) are the fifth heaviest extant land mammals (*vs.* three species of elephants, and white rhinoceros) albeit close in size to black rhinoceros (references); at ~1,500 kg adult body mass (*Owen-Smith, 1988*; *Silva & Downing, 1995*), with the largest recorded individuals surpassing 3,800 kg (cited in *Garland, 1983*). At such large sizes, they offer important comparative information on how body size influences locomotor abilities on land. In general, mammals >100 kg are able to move more slowly than medium-sized mammals, and at extreme sizes >1,000 kg, maximal speed and even gait capacities decrease (*Garland, 1983*; *Dick & Clemente, 2017*; *Hutchinson, 2021*). For example, elephants are well known to not move faster than about 7 ms$^{-1}$ (25 kph), retaining a lateral sequence symmetrical footfall pattern (*i.e.*, foot contact sequence: left hind, left fore, right hind, right fore; *Muybridge, 1887*; *Slijper, 1946*; *Cartmill, Lemelin & Schmitt, 2002*; *Struble & Gibb, 2022*) never with an aerial (suspended) phase

Corresponding author
John R. Hutchinson,
jrhutch@rvc.ac.uk

(*e.g.*, *Gambaryan, 1974*; *Hutchinson et al., 2003*, *2006*). The largest extant rhinoceroses might not move much faster than elephants, although reliable data are very scarce for them–yet they can use an asymmetrical transverse galloping gait with an aerial phase (*Slijper, 1946*; *Dagg, 1973*; *Gambaryan, 1974*; *Hildebrand, 1977*; *Alexander & Pond, 1992*; *Paul & Christiansen, 2000*). Giraffes, which are substantially smaller (<1,300 kg), use a lateral sequence walk and rotary gallop; not trotting (*e.g.*, *Hildebrand, 1965*, *1980*, *Dagg, 1973*, *1979*; *Gambaryan, 1974*; *Hildebrand, 1977*; *Basu, Wilson & Hutchinson, 2019*; *Basu et al., 2019*).

Similarly, there are few empirical data for terrestrial locomotion in the Common hippopotamus or their relatives the pygmy hippopotamus (*Choeropsis liberiensis* or *Hexaprotodon liberiensis*). Maximal speeds have not been reliably measured but have been claimed (without empirical documentation) at up to ~8 ms$^{-1}$ (~30 kph; *Dagg, 1973*; *Bakker, 1975*; *Garland, 1983*; *Kingdon, 1989*; *Nowak, 1999*). *Hildebrand (1989)* provided some footfall pattern data for the pygmy hippopotamus, indicating a lateral sequence walk, but noted that Common hippopotamus (onwards here, simply "hippos") will use a walking trot (*Hildebrand, 1962*, *1965*, *1967*, *1976*) and running trot (*Hildebrand, 1980*, *1989*); meaning a symmetrical gait with diagonally synchronised limbs. *Dagg (1973*; also *Howell, 1944)* gave a basic classification of gaits in hippos, listing them as using a typical (lateral sequence) walk, but no indication of a 'running walk', and also as using a trot; but no galloping gait (stating they "cannot hoist themselves into the air"). They commented that prior studies such as *Slijper (1946)* classified a faster hippo gait as an 'amble' (lateral sequence), but that this term was too "ambiguous", implying it did not apply well to hippos. *Dagg (1979)* followed up with analysis of video data for 29 strides of a hippo that "did not use lateral supporting legs at all", concluding that the wide body and relatively short legs (*e.g.*, *Bakker, 1975*) prevent balance on two ipsilateral legs (presumably due to excessive rolling momentum). Other studies of slower walking locomotion in hippos have observed lateral sequence footfall patterns (*Niemitz, 2001*; also see *Coughlin & Fish, 2009*), albeit sometimes closer to diagonal sequence (*e.g.*, *Catavitello, Ivanenko & Lacquaniti, 2018*). *Usherwood & Self Davies (2017)* obtained data for hippos using slow trotting, suggesting that this pattern minimises mechanical and perhaps metabolic energy usage (also see *Smith & Usherwood, 2020*). In contrast, *Coughlin & Fish (2009)* showed that hippos can use asymmetrical "punting" locomotion underwater (also see *Niemitz, 2001*; *Bennett, Morse & Falkingham, 2014*; *Mazza, 2014*; *Van der Geer, Anastasakis & Lyras, 2015*; *contra Dagg, 1973*). While hippos spend much time in the water or resting (~50%), their locomotion on land must remain important because they still spend around one-third of their time doing so, particularly at night (*e.g.*, *Owen-Smith, 1988*; *Timbuka, 2012*; *Mekonen & Hailemariam, 2016*; *Fernandez, Ramirez & Hawkes, 2020*).

It has become at least tacitly accepted that hippos do not use gaits on land that are more extreme than somewhat fast symmetrical trotting (*i.e.*, no asymmetrical cantering or galloping), but it is unclear if hippos use aerial (suspended) phases at any speeds or gaits (*e.g.*, *Christiansen & Paul, 2001*; *Thomson, 2019*; *Hutchinson, 2021*). Furthermore, even the basic walking patterns of hippos are almost uncharacterised; do they only use slower lateral sequence walking and then switch to faster trotting footfall patterns? Here, to address these

uncertainties and to further understanding of hippo locomotion, we use video analysis to quantify their footfall patterns and stride kinematics on land.

## MATERIALS AND METHODS

We compiled basic kinematic data from videos of locomoting hippos from two sources: first, from an internet search (not exhaustive, but particularly searching for faster running-type behaviours), and second, from firsthand data collection. Each video was split into one or more "trials", with a trial defined as a cyclic sequence of one or more strides from the same individual. The first source of data is listed in the Supplemental Information. Empirical data for the second source were obtained at Flamingo Land Resort (Kirby Misperton, North Yorkshire, UK), as follows. We had access to two adult hippos: a 19 year old male (hip height ~1.3 m) and 23 year old female (hip height ~1.2 m). Body masses were not known for any subjects. At Flamingo Land Resort, a GoPro Hero 3+ (San Mateo, CA, USA) camera was set on a tripod and positioned along the fence of the hippos' yard. This placement ensured the hippos would walk roughly parallel to the camera's field of view. The camera was switched on (recording at 30 Hz, $1,920 \times 1,080$ pixel resolution) and left running for when the hippos came out of the water and walked across the yard to their stable. They were also encouraged to move freely in front of the camera by placing food outside. A total of 16 h of footage across 2 days was recorded in this manner. It was difficult to encourage the hippos to move, because it was not desirable to cause the hippos any stress or use unethical stimuli to cause them to run. Thus we captured only the walking data that they were willing to provide. Ethics approval was given by the Clinical Research Ethical Review Board at the Royal Veterinary College; approval number CR2022-003-2.

GoPro light videos then were uploaded onto a laptop into GoPro Studio software (GoPro, San Mateo, CA, USA), where "fisheye" effects were removed, and the videos were converted to AVI format. All videos were inspected and trials which included obvious non-steady locomotion such as acceleration, deceleration, turning, or slow grazing were discarded. Each video was analysed frame by frame in VirtualDub software (https://virtualdub.sourceforge.net/) and the times at which each foot hit and left the ground (as long as feet approximately were visible) were estimated. "Foot-on" was defined as the frame that the limb ceased travelling forward; "foot-off" was defined as the frame that the limb began travelling forwards/upwards; see Fig. 1. These definitions helped to reduce errors caused by blurriness, rough terrain, obscured feet, or feet brushing the ground. Footfall patterns and other basic stride kinematics were determined from these timing data, as follows.

We followed the standard approach of defining each limb's phase as the fraction of a stride relative to the left hind foot contact (=0.0). Biknevicius & Reilly (2006) suggested that trotting can have (left fore and right hind) limb phases of between ~0.44 and 0.56 (0.50 ± 0.0625), encompassing potential variation; we initially employ that definition here (and right fore phases should be ~0.94–1.0). Their set of footfall definitions encompasses eight primary footfall patterns (trot, pace, lateral sequence and diagonal sequence; and two couplet varieties each for the latter two; following the work of Hildebrand and others as per the Introduction). However, they concluded that narrowing this to four
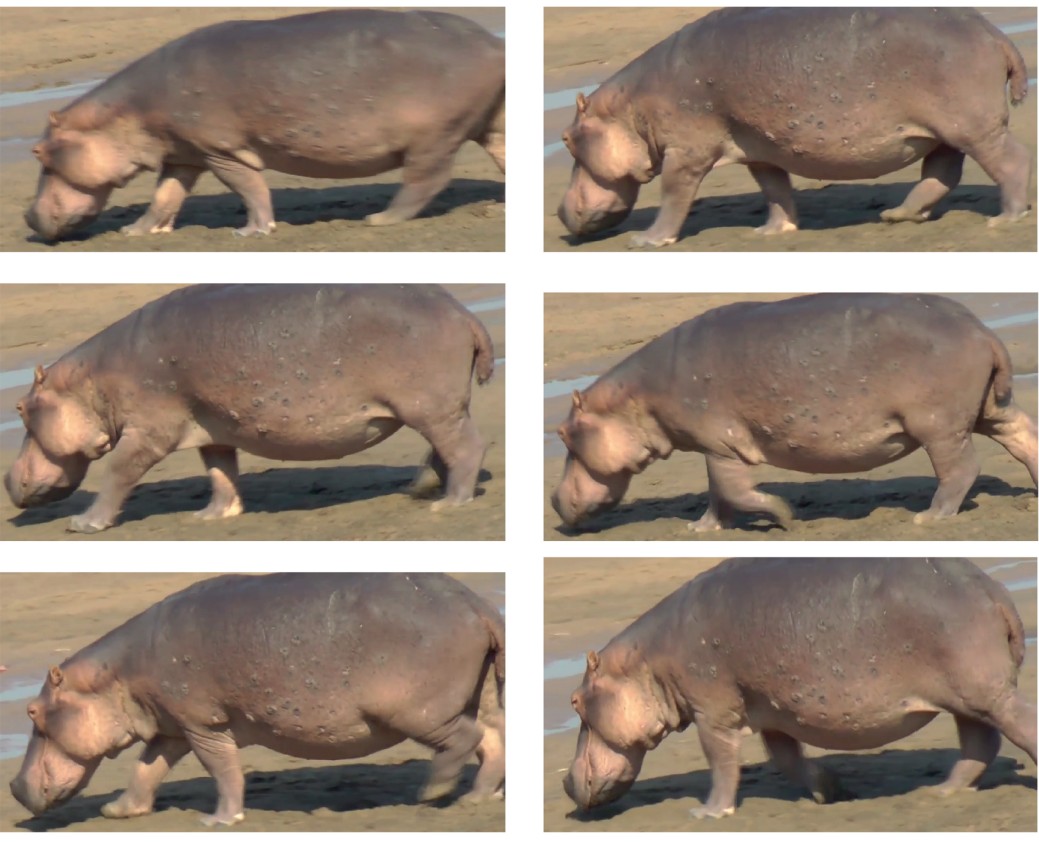

**Figure 1 Footfall patterns and foot-on/foot-off definition examples for hippo locomotion data analysis.** Sequence from left to right, top to bottom is: left hind foot-on, right hind foot-on, left front foot-on, left front foot-off, right front foot-on, right front foot-off. Here, the footfall pattern deviates slightly from an ideal trot, but the foot contact sequence still is (left hind, right fore), (left fore, left hind). Images captured and cropped from internet video at: https://www.youtube.com/watch?v=ej6DrFpnJrk. Image credit: Victoria Wallace, Director, Zikomosafari.

(omitting couplets) primary patterns is more pragmatic. That re-definition would allow phase variation to fall within ±0.125 of a stride, so we consider this issue in assessing how our limb phase results compare with definitions of primary patterns. We also quantified duty factor (DF = ground contact time; *i.e.*, stance phase duration divided by stride duration) and stride frequency (SF = 1/stride duration). Here we loosely term gaits with DF ≥ 0.5 as walking and DF < 0.5 as running. As none of our videos had reliable scale objects for calibrating distances, we could not calculate velocity-related parameters including stride lengths. This was because distances were unknown in the internet videos, and we were unable to place a scale object in the hippo enclosure during our filming. However, it provided the advantage that we could obtain more trials because we did not have to rely on strides and trials that were parallel to the camera's field of view and in the same plane as a scale object.

Our internet search for videos initially yielded 25 videos containing 34 trials used here. *Usherwood & Self Davies (2017)* obtained kinematic data from five videos (one no longer available; all obtained *via* www.youtube.com) for hippos. Similarly, *Lees et al. (2016)*

measured (again from www.youtube.com; three videos but one redundant with our sample) hippos walking. We reanalysed these seven videos here as part of our final sample of 23 videos. Our data collection at Flamingo Land Resort provided 12 valid trials. All trials were composed of between one and eight continuous strides that we used for final data analyses; ultimately totalling 169 strides from the 46 trials for a total of 32 individual hippos. Details on videos and trials can be found in Table S1.

Our study's aims are descriptive, so we only used basic statistics. We plotted limb phases (LF = left front; RF = right fore; RH = right hind) and stride frequencies (SF) against duty factor (DF; which should be roughly inversely proportional to speed) using regressions in GraphPad Prism 10 software (Boston, MA, USA). We also calculated stance and swing phase durations. In analysing data, we took the mean values of each of these stride parameters for each trial.

## RESULTS

On average, the hippos we analysed were trotting: mean limb phases were RH = 0.51 ± SD 0.051, RF = 0.94 ± SD 0.040 and LF = 0.46 ± SD 0.050. No limb phases showed significant changes with DF (Fig. 2: linear regressions: $p$ values for RH, RF and LF = 0.0925, 0.155 and 0.153) and all phases had appreciable variation ($R^2$ for RH, RF and LF = 0.063, 0.045, 0.046). This finding suggests that the hippos typically were demonstrating a 'walking' trot at DF ≥ 0.5 and a 'running' trot at DF < 0.5. Some trials marginally varied into more lateral or diagonal sequence patterns overall (Fig. 2; see Discussion). The Supplemental videos (available at https://figshare.com/s/99e958f4094b1cc24e80; doi:10.6084/m9.figshare.25027142) show trials for the two hippos at Flamingo Land Resort.

Stance durations ranged from 0.17–5.48 s (the latter a very slow value for a slowly grazing hippo); and swing durations from 0.13–0.58 s (Fig. 3). SF decreased with DF, and ranged from 0.17–3.2 (the highest values 2.8 and 3.2 Hz in fighting hippos) (Fig. 4). Mean DF varied between 0.36–0.94. Forelimb $vs.$ hindlimb DFs showed high variation (means 0.66 ± SD 0.18; 0.65 ± SD 0.17) but forelimbs had slightly (median 0.01) greater DFs than hindlimbs (median 0.76 $vs.$ 0.75; Wilcoxon matched-pairs signed rank test, two-tailed: $p$ = 0.0041, $n$ = 46). Overall, it seems that during hippo locomotion studied, DF was decreased by decreasing mean stance durations ~32 times and swing durations only ~4.5 times (Fig. 3), thereby increasing SF ~19x (Fig. 4) while DF decreased to approximately one-third slow-walking values (i.e., roughly, speed is increased more by decreasing stance; not so much swing; durations).

As the figures show (see also frequency distribution in Fig. S1), we obtained two main clusters of data, corresponding to mean DF ~0.36–0.58 and ~0.75–0.94. This clustering was largely caused by our sourcing of 'running' videos from the internet, and then walking videos from our Flamingo Land sample plus some more videos from the internet. It is unclear if the apparent gap between DF ~0.58–0.75 is artefactual based on this sample, or reflective of either gait transition or preferred speeds.

Importantly, the 12 'running' trot (mean DF < 0.5) videos obtained from the internet appeared to (and in some cases, clearly showed) brief aerial phases (e.g., Fig. 5), although some footfall patterns deviating slightly from an ideal trot seemed to prevent aerial

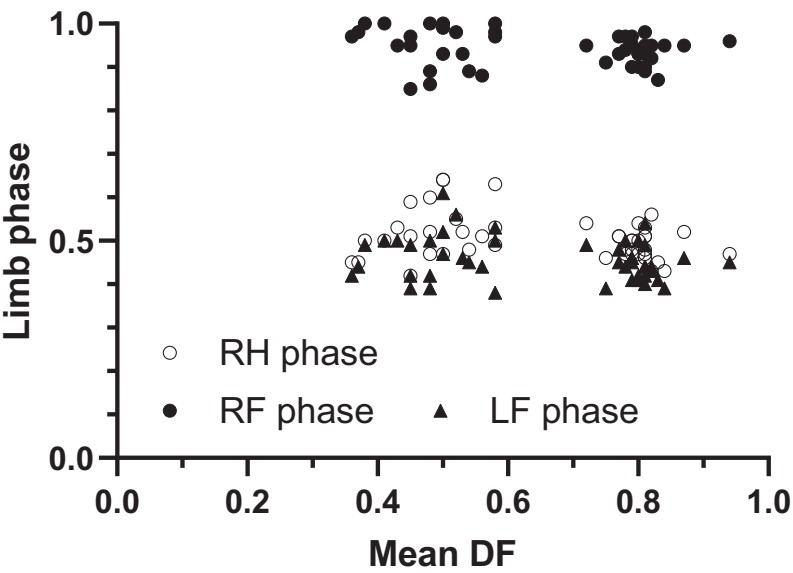

**Figure 2 Hippos' limb phases plotted *vs*. mean duty factor (DF).** RH = right hind; RF = right front; LF = left front. All phases are the fraction of a stride for foot contact timing *vs*. left hind's.

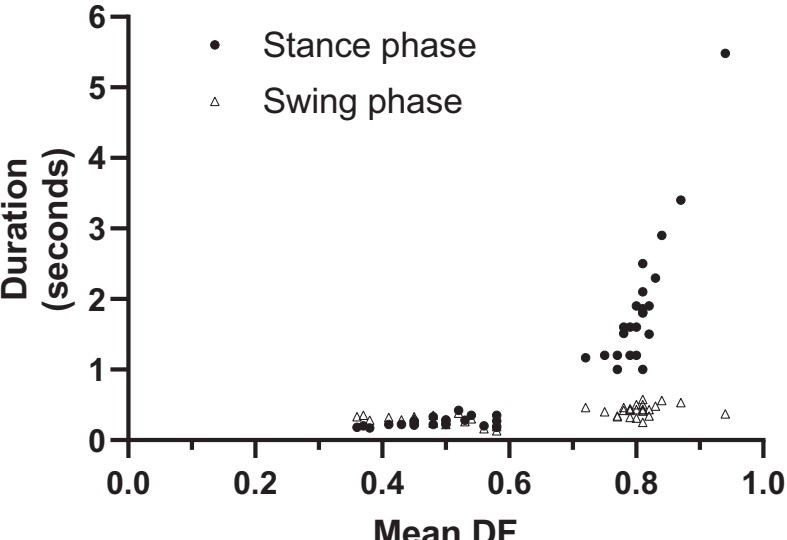

**Figure 3 Hippos' stance and swing phase durations plotted *vs*. mean duty factor (DF).** Equations: stance duration = $8.506\ \text{DF}^{7.188}$ ($R^2 = 0.934$); swing duration = $0.4817\ \text{DF}^{0.7119}$ ($R^2 = 0.391$); $n = 46$ and df = 44.

phases. As per *Hutchinson et al. (2006)*, an aerial phase would occur if the DF was less than the greatest phase difference between two consecutive footfalls (always ipsilateral limbs). This difference was a mean of 0.54 for LH *vs*. LF (= 1−0.46) and 0.43 for RF *vs*. RH (= 0.94−0.51), which is roughly 0.5 overall, so trots with DF < 0.5 would be expected to involve aerial phases, which they did.

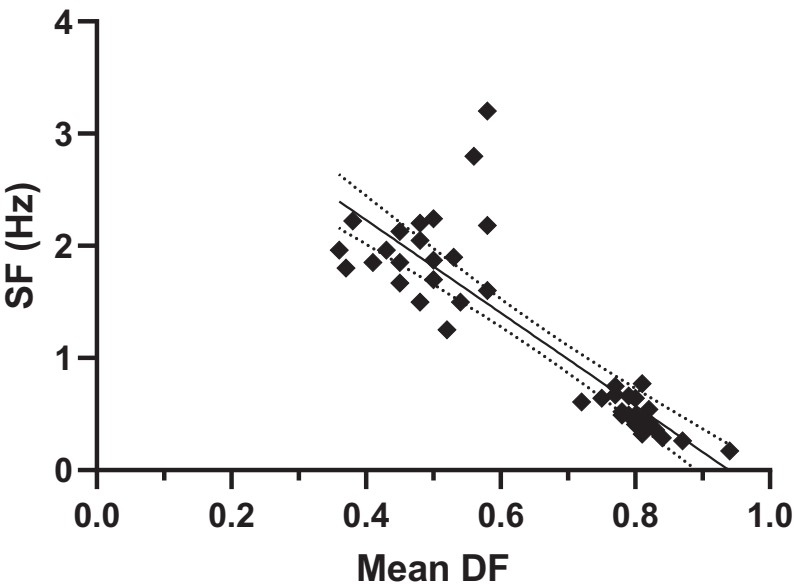

**Figure 4  Hippos' stride frequency (SF) plotted *vs.* mean duty factor (DF).** Equation: SF = −4.14 DF + 3.89; $p < 0.0001$; $R^2 = 0.763$.

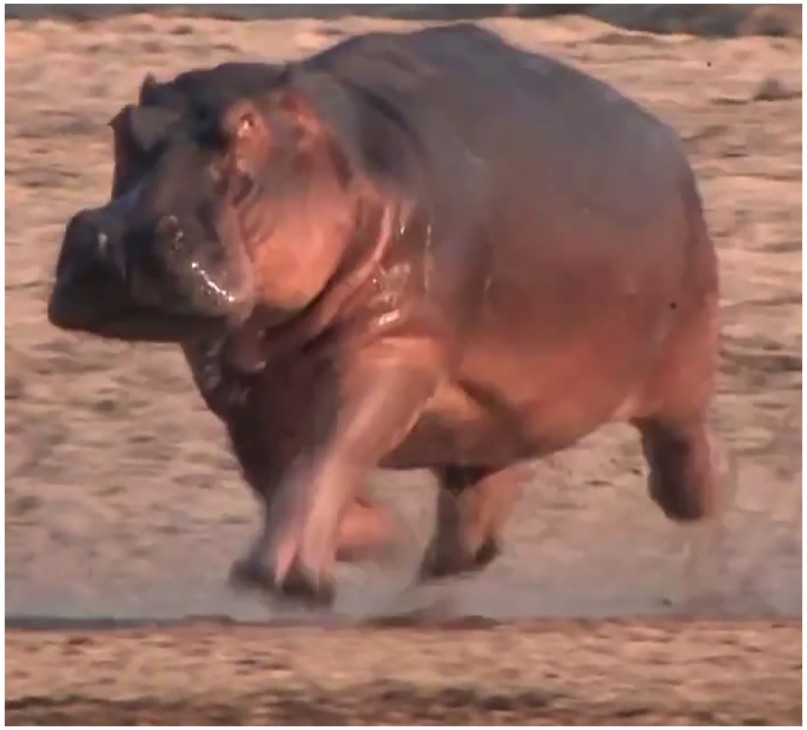

**Figure 5  Example of an aerial phase in a hippo.** From internet video: https://www.shutterstock.com/video/clip-1015076347-hippos-chasing-each-other-riverbed, used with permission. Image credit: Johan Vermeulen.

## DISCUSSION

Our most notable findings are that hippos normally trot (whether walking or running), and do use an aerial phase when running quickly; the latter to our knowledge has not been reported before. Interestingly, hippos studied did not normally change their footfall pattern across gaits; instead they used one main spectrum spanning from a 'walking' trot to a 'running' trot (Figs. 1, 2, 5). This is an unusual pattern for large terrestrial mammals: elephant use lateral sequence gaits; rhinoceroses (as far as is known, for all extant species) use a lateral sequence walk, true trot and transverse gallop; giraffes use a lateral sequence walk and a rotary gallop (see Introduction). Compared with smaller mammals, the gait repertoire of rhinos (and to a degree, giraffes) is more "normal" (and ancestral, evolutionarily, for most mammals), raising the question of why hippos trot and never gallop on land–but have not entirely lost the ability to use an aerial phase. Two not mutually exclusive hypotheses are that hippos are too large; with low strength: weight ratios; and too aquatically adapted for athletic locomotion on land. However, if hippos can indeed obtain speeds of ~7 ms$^{-1}$, they would match the performance of elephants, and their lowest DF observed here (0.36) is almost identical to known DFs of elephants of any size or species (minimal observed DF 0.37; *Hutchinson et al., 2003*, *2006*). Intriguingly, elephants also have evolved with some aquatic adaptations as members of the clade Tethytheria (*e.g.*, *Gaeth, Short & Renfree, 1999*), so the evolutionary and biomechanical explanations for hippo and elephant maximal locomotor performance may be similar to some degree. The maintenance of lateral sequence (not trotting) footfall patterns across all DFs in elephants helps explain why, kinematically, they do not use aerial phases at DFs similar to those of fast-running hippos, because that footfall pattern evenly spaces out ground contacts whereas trotting involves (near-) synchronous limb pairs.

Our fastest hippos were using extreme behaviours (in the wild or in captivity) that may be near-maximal performance: avoiding lions or rhinoceroses or aggressive behaviour toward each other or humans and their vehicles. Maximal measured athletic capacities for stride parameters for elephants *vs.* hippos are: in elephants, maximal SF 1.9 (hippos = 3.2) and minimal stance and swing durations of 0.20 (hippos = 0.17) and 0.28 (hippos = 0.13) s (*Hutchinson et al., 2003*, *2006*). Unlike elephants, hippos appear to change speed mainly by decreasing stance duration (and thus DF; and increasing SF) rather than swing duration. Thus overall, while it remains unclear if hippos can achieve faster speeds than elephants do, they clearly reach greater SFs and smaller DFs and swing durations (with slightly smaller stance durations), and thus show relatively greater athleticism.

Figure 2 (see also Fig. 1; Table S1; Supplemental videos) shows some variation for some limbs' phases that does not correspond to ideal trotting patterns. We suspect that the variation in RF and LF phases observed; which is similar to the variation for LH phases; arises from the high prevalence of non-steady locomotion in our data, imposed by the variety of the sources used and the substrates and other conditions of filming those individuals. Indeed, if lateral or diagonal sequence walking patterns were common in hippos, we would expect to see clusters of RF and LF phases that were both distinct from RH ~0.50 and LH ~0.00. The variation we observed does not indicate distinct, consistent

"singlefoot" lateral or diagonal sequence walking, although some trials or strides seem close to having isolated lateral or diagonal couplets or "dirty trots" (*sensu Biknevicius & Reilly, 2006* and references therein). This issue is more of a concern if strictly using the eight primary footfall pattern categories (separated by 0.0625 phase boundaries) rather than the simpler four (separated by 0.125 phase boundaries) promoted by *Biknevicius & Reilly (2006)*. Considering the observed variation and the semantics of definitions, we see the simplest message of our data is that hippos trot.

The absence of clear lateral sequence walks in our data, *vs.* several claims that these exist in hippos (*Slijper, 1946*; *Dagg, 1973*; *Niemitz, 2001*; *Coughlin & Fish, 2009*; *Catavitello, Ivanenko & Lacquaniti, 2018*), is perplexing, but strongly concurs with numerous prior studies, including quantitative ones (see Introduction). Lateral sequences might be used for some very slow walking (*e.g.*, while grazing); as *Hildebrand (1976)* intimated; yet we did not observe strong examples of that pattern in our slow walking trials. Compared with Common hippopotamuses, pygmy hippopotamuses spend less time underwater and more time on land (*Walker, 2008*). This might explain why *Hildebrand (1989)* found that they used more lateral sequence walking, but the circumstances under which he collected both species' data remain unknown; the differences might purely be variation. *Usherwood & Self Davies (2017)* analysed five videos of hippos using slow (DF 0.76) trotting (RH and LF limb phases ~0.45), in agreement with our findings. *Lees et al. (2016)* studied three videos that we also find to involve slow trotting, and with stance and swing durations of 1.12–1.48 and 0.41–0.52 s, DF 0.73–0.74 and SF 0.50–0.65; which our re-analyses of these data concur with. *Dagg (1979)* studied hippos with DF ~0.79. All of these published data fit within variation of our other trials (Figs. 2–4).

We found small (~0.01) differences in hippo forelimb and hindlimb DFs but we deem these probably to not be of much biomechanical significance. However, the expectedly greater antigravity supportive roles of the forelimbs (typical of quadrupedal mammals; *e.g.*, *Basu, Wilson & Hutchinson, 2019*) would benefit from greater forelimb DFs to moderate ground reaction forces. Considering the lowest mean DF of 0.36, we would expect one forelimb ~DF 0.37 to involve a peak vertical ground reaction force of about 1.27 times body weight (see *Alexander & Pond, 1992* for example of this method applied to a rhino). This is slightly greater than the force of 1.1 times body weight measured and extrapolated for an elephant's forelimb (*Ren et al., 2010*), but presumes an ideal half cosine cycle of vertical force *vs.* time.

Our study was limited by inability to measure velocity-related parameters, which would be particularly valuable to have for fast-moving hippos. Furthermore, no kinetic data exist for hippos. Such data would reveal limb forces and mechanical energy fluctuations; the latter being crucial for testing biomechanical gait transitions (*i.e.*, vaulting to bouncing gaits; see *Biknevicius & Reilly, 2006*; *Struble & Gibb, 2022*). Almost all of our data involved 25–30 Hz videos, which inevitably would introduce more imprecision and noise into the fastest trials (*e.g.*, one frame at 0.04 s is ~31% of swing phase duration at 0.13 s). Thus surely some of the variation in our results is due to human error and resolution of hardware; and absence of quantitative data on steady-state locomotion. Similarly, no hippos were moving over ideally smooth, level terrain and few were moving in straight

forward directions. However, some of these problems could be seen as benefits, too. Our measurements are for hippos either in their natural environments (most of the internet videos used) or else in fairly natural terrain in captivity. Thus the kinematic patterns we measured should better reflect normal locomotion overall. Our study's decent sample size of 169 strides from 46 trials of 32 individual hippos, across a broad DF (or, qualitatively, speed) range defends its value. Unfortunately all of our fastest (mean DF < 0.50) trials had video footage involving extensive camera panning, zooming and/or direction of motion out of the plane of the video, so it was not feasible to quantify maximal relative speeds of hippos in body lengths per second ($s^{-1}$) for comparative analysis (see *Iriarte-Díaz, 2002*).

Our data and conclusions are relevant to clinical veterinary care of hippos, especially detection of lameness *via* comparison of an animal's gait to a "normal" patterns (*e.g.*, *Hilsberg-Merz, 2008*; *Ren et al., 2010*; *Dadone, 2018*; *Panagiotopoulou, Pataky & Hutchinson, 2019*; *Turner et al., 2023*). *Flacke et al. (2016)* analysed 43% of all pygmy hippos that had died in captivity and concluded that musculoskeletal or neuromuscular degenerative diseases (*e.g.*, *Hittmair & Vielgrader, 2000*; *Regnault et al., 2013*; *Jones, Gasper & Mitchell, 2018*; *Dadone et al., 2019*) were common or the most common causes for euthanasia of adult or geriatric individuals. These hippos likely would have presented with clinical signs of lameness. Improved detection of lameness would aid identification and monitoring of hippo foot/limb pathology and development of treatments such as appropriate flooring of enclosures or ideal environments for encouraging exercise in captive animals.

New information on hippo locomotion that we provide here is also useful for understanding the evolution of locomotion, body size, habitat usage and ecology in Hippopotamidae (*e.g.*, *Sondaar, 1977*; *Boisserie et al., 2011*; *Mazza, 2014*; *Van der Geer, Anastasakis & Lyras, 2015*; *Rozzi et al., 2020*; *Houssaye et al., 2021*; *Georgitsis et al., 2022*). It is conspicuous that there is one recorded instance of a young pygmy hippopotamus at Tampas Lowry Park Zoo (Florida) using a transverse gallop (https://www.youtube.com/watch?v=nmsZwwWYJkE; includes an adult quickly trotting) which could have important implications for the biomechanics of hippopotamuses across ontogeny and size variation, and through evolution. A fascinating question is how (and when) hippos seem to have lost their ability to use asymmetrical gaits and now only trot? Also, how tightly were changes in athleticism linked to aquatic habitats, body size and morphology in different hippopotamid lineages? This is an excellent example of where integration of biomechanics and evolution could be powerful and exciting.

## ACKNOWLEDGEMENTS

We thank Alexandros Stefanidis, Sharon Warner and members of the Structure and Motion Laboratory for supporting experimental data collection and analysis. We thank Kieran Holliday and staff at Flamingo Land Resort for supporting our data collection, and Guy Leahy for noting the galloping pygmy hippopotamus video. We are grateful for constructive peer reviews from Christofer Clemente and an anonymous referee.

### Funding

This work was supported by the Royal Veterinary College. The funders had no role in study design, data collection and analysis, decision to publish, or preparation of the manuscript.

### Grant Disclosures

The following grant information was disclosed by the authors:
Royal Veterinary College.

### Competing Interests

John Hutchinson is an Academic Editor for PeerJ.

### Author Contributions

- John R. Hutchinson conceived and designed the experiments, analyzed the data, prepared figures and/or tables, authored or reviewed drafts of the article, and approved the final draft.
- Emily V. Pringle conceived and designed the experiments, performed the experiments, analyzed the data, authored or reviewed drafts of the article, and approved the final draft.

### Animal Ethics

The following information was supplied relating to ethical approvals (*i.e.*, approving body and any reference numbers):

The Clinical Research Ethical Review Board at the Royal Veterinary College approved the study (Approval number CR2022-003-2).

### Data Availability

Raw measurements are available in the Supplemental Files.

Videos showing the original data analysed are available at Figshare:

Hutchinson, John (2024). Supplementary Videos: walking hippos. figshare. Dataset. https://doi.org/10.6084/m9.figshare.25027142.v1.

### Supplemental Information

Supplemental information for this article can be found online at http://dx.doi.org/10.7717/peerj.17675#supplemental-information.

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
