# Peer review of "Footfall patterns and stride parameters of Common hippopotamus (Hippopotamus amphibius) on land"

_PeerJ, doi:10.7717/peerj.17675_

## Round 0.1 · original submission · Minor Revisions

The referees between them raised very few points but I think these are reasonable and should be addressed in the second submission, but this is clearly a very good paper with relatively little to attend to in order to make it publishable.

·

Basic reporting

Well written, very clear.

Although the paper has a link to a figshare - this appears to be directed towards a video. There was no link given in the paper directed toward the raw data.

Experimental design

The experimental design was simple but effective for this study. I wonder if some more information could be determined from these videos. For example, what proportion of videos were walking perpendicular to the FOV of the camera? There are likely some and if so could the speed of movement and stride length not be estimated in units of body lengths per second?
Previous studies have used this metric for comparison with other species, so it might be useful for further researchers?

Iriarte-Díaz, J., 2002. Differential scaling of locomotor performance in small and large terrestrial mammals. Journal of Experimental Biology, 205(18), pp.2897-2908.

Validity of the findings

Data is overall presented well, however a few key points were missing, or could improve the manuscript with their addition.
1- Why were there two obvious groups of data (large and small duty factors?) - was this by design? I think this needs to be clearly addressed in the results.
2- I would like to see a frequency distribution of all ~160 strides presented. This might be useful for future studies who might want to extract the preferred stride frequency of walking hippos.
3- Given you have contact times, it might be instructive to estimate peak vertical ground reaction forces (e.g. by the simple calculations in Alexander etc). As above this might be a nice bit of data which could be used in future studies in a broad comparative sense.

Additional comments

Overall, this is a very nice and simple paper. I feel more papers like this should be published. I like that the data is clearly represented.

Reviewer 2 ·

Basic reporting

Dear Editor,

I have considered the manuscript by Hutchinson and Pringle on ‘Footfall patterns and stride parameters of Common hippopotamus (Hippopotamus amphibius) on land’. Although this paper is highly descriptive (as also acknowledged by the authors), I believe this contribution is highly valuable for researchers working in the fields of evolutionary and functional morphology, locomotor behavior and biomechanics and, as mentioned by the authors, it might also be important for animal welfare when helping zoo veterinarians in evaluating gaits for lameness.

The paper is very well written and thus also accessible for the less specialized reader. I do appreciate the comprehensive literature review (including also the ‘older’ literature). The authors are also well aware of the limitations of the methods used (no speed related info can be extracted from the videos used), but manage to deal with ‘speed relations’ in a most meaningful way (also well explained; but see further). The discussion is ‘to-the-point’ and especially the interpretation of the comparison with the available data of elephant locomotion (the first author is definitely a specialist on the topic) is revealing.
As a result of the descriptive nature (and the clear writing style), I have few questions or comments. I have one suggestion and one question.

Although it was mentioned that speed related info is missing, the results repeatedly refers to relationships of phases and frequency etc. with speed. In the Materials & Methods section (Line 137-138) it is mentioned that these regressions are against DF (mentioning only marginally ‘which should be roughly inversely proportional to speed’). Personally, I would spend a few more words on this, in order to avoid potential confusion when reading results and discussion (e.g. mentioning explicitly that each time ‘speed’ (in italics) is used, this actually means 1/DF). Also for the figures, it could be helpful to have the dimensions of the X-axes as 1/DF and plotting these in an inverse order (so that lowest ‘speeds’ show up at the left hand sides of the graph; highest ‘speeds at the righthand sides). For one or another reason, I did not find figure legends (I got a word version of the manuscript and separate files for the illustrations). I assume it is not intentional to have no legends?

In line 148 it is mentioned ‘Some trials marginally varied into more lateral or diagonal sequence patterns overall’. From the video in supplementary material the deviation from the trotting pattern is rather conspicuous (at the start of the sequence for the individual entering the stable first and just before the second individual enters the stable). Why is this considered ‘marginal’? Also from fig.2 it is obvious that several strides are clearly outside the 0.5+-0.06 range? Related to this, the statement in line 206 of the discussion seems to conflict with what can be seen in the example video.

Experimental design

no comments

Validity of the findings

no comments

Additional comments

see first box

---

## Round 0.2 · accepted · Accept

The original comments from the referees were very minor and I think they have all been addresses adequately, so I'm happy to accept this paper.